# Biology and Development of DNA-Targeted Drugs, Focusing on Synthetic Lethality, DNA Repair, and Epigenetic Modifications for Cancer: A Review

**DOI:** 10.3390/ijms25020752

**Published:** 2024-01-06

**Authors:** Kiyotaka Watanabe, Nobuhiko Seki

**Affiliations:** Department of Medicine, School of Medicine, Teikyo University, 2-11-1 Kaga, Itabashi-ku, Tokyo 173-8605, Japan

**Keywords:** DNA-targeted drugs, targeted therapy, next generation sequencing, PARP inhibitors, SLFN11, MGMT, ATR kinase, epigenetic modification

## Abstract

DNA-targeted drugs constitute a specialized category of pharmaceuticals developed for cancer treatment, directly influencing various cellular processes involving DNA. These drugs aim to enhance treatment efficacy and minimize side effects by specifically targeting molecules or pathways crucial to cancer growth. Unlike conventional chemotherapeutic drugs, recent discoveries have yielded DNA-targeted agents with improved effectiveness, and a new generation is anticipated to be even more specific and potent. The sequencing of the human genome in 2001 marked a transformative milestone, contributing significantly to the advancement of targeted therapy and precision medicine. Anticipated progress in precision medicine is closely tied to the continuous development in the exploration of synthetic lethality, DNA repair, and expression regulatory mechanisms, including epigenetic modifications. The integration of technologies like circulating tumor DNA (ctDNA) analysis further enhances our ability to elucidate crucial regulatory factors, promising a more effective era of precision medicine. The combination of genomic knowledge and technological progress has led to a surge in clinical trials focusing on precision medicine. These trials utilize biomarkers for identifying genetic alterations, molecular profiling for potential therapeutic targets, and tailored cancer treatments addressing multiple genetic changes. The evolving landscape of genomics has prompted a paradigm shift from tumor-centric to individualized, genome-directed treatments based on biomarker analysis for each patient. The current treatment strategy involves identifying target genes or pathways, exploring drugs affecting these targets, and predicting adverse events. This review highlights strategies incorporating DNA-targeted drugs, such as PARP inhibitors, SLFN11, methylguanine methyltransferase (MGMT), and ATR kinase.

## 1. Introduction

DNA-targeted drugs are a class of pharmaceutical agents designed to interact with and modulate the activity of DNA molecules within cells. These drugs are developed to treat various medical conditions, including cancer, genetic disorders, and infectious diseases, by directly impacting DNA structure, replication, transcription, repair, or other essential cellular processes involving DNA. DNA-targeting in cancer treatment aims to improve the effectiveness of treatment and minimize side effects, as it targets specific molecules or pathways involved in cancer growth and progression.

The inception of DNA-targeting cancer drugs traces back to Watson and Crick’s 1953 DNA structure revelation. The foundational groundwork was established by early chemotherapy, including nitrogen mustards. Progress in DNA replication and repair paved the way for drugs like etoposide in the 1970s.

Many drugs used in cancer treatments target DNA at the molecular level, considering it a non-specific target for cytotoxic agents. While this holds true for conventional chemotherapeutic drugs, newer agents discovered in recent times exhibit improved effectiveness [1]. Furthermore, genomic insights and poly-ADP-ribose polymerase (PARP) inhibitors (2010s) exemplify evolving precision medicine. A new generation of DNA-targeted drugs exploring synthetic lethality, DNA repair, and expression regulatory mechanisms, including epigenetic modifications, is anticipated to be far more specific and effective.

## 2. Understanding DNA Structure and Function

DNA is a complex molecule that carries genetic information in living organisms. It consists of two long strands arranged in a double helix, with each strand composed of nucleotides containing a sugar, a phosphate group, and one of four nitrogenous bases: adenine (A), thymine (T), cytosine (C), and guanine (G). Complementary base pairing (A-T and C-G) allows DNA to replicate and transmit genetic information accurately during cell division. DNA-targeted drugs exert their effects through various mechanisms, such as intercalation, alkylation, topoisomerase inhibition, and DNA cross-linking.

The sequencing of the human genome in 2001 significantly contributed to the advancement of targeted therapy and precision medicine [2]. This milestone marked a profound understanding of the genetic structure of the human body, enabling the identification of precise genetic mutations and variations responsible for the emergence and evolution of various diseases, including cancer. The combination of knowledge about the human genome and technological progress has spurred a proliferation of clinical trials in precision medicine in recent years. In these trials, the utilization of various biomarkers aids in the identification of specific genetic alterations; molecular profiling identifies potential therapeutic targets, and cancer treatments are tailored to address multiple genetic changes [3,4].

## 3. Characterizing Cancers to Targeted Treatments

Next-generation sequencing (NGS) has revealed that genomic alterations in advanced cancers deviate from the conventional categories established by the organ of tumor origin. In addition, NGS has brought to light the unique and complex genomic and immune profiles of metastatic tumors, highlighting the importance of tailoring treatments based on genomic analysis [5,6]. Recent technological advancements have substantially reduced the cost and time associated with sequencing, rendering it more accessible for researchers to undertake such studies and for clinicians to apply the information in treating cancer.

Circulating tumor DNA testing, known as ctDNA, represents a further advancement in cancer characterization. This non-invasive method is progressively employed to choose drugs and evaluate the response to treatment. As ctDNA captures the genetic characteristics of the entire tumor and not just a single biopsy sample, it delivers a more thorough understanding of the tumor’s genetic diversity. The process of ctDNA testing relies on detecting DNA that has been released into the bloodstream from the patient’s tumor. This is particularly advantageous in cases where the tumor exhibits multiple subclones [7]. ctDNA testing serves as a valuable tool for monitoring the treatment response, capable of identifying alterations in the levels of ctDNA in the bloodstream during the course of treatment. It offers significant information about the dynamics of subclones within the tumor, guiding decisions on therapeutic interventions [8]. In ctDNA analysis, the detection of *BRCA1*/*2* mutations can guide treatment with PARP inhibitors, and the detection of DNA methylation of selected tumor suppressor genes can provide prognostic information in various types of cancer.

## 4. Precision Medicine and Molecular Markers

Examples of effective targeted therapies tailored to molecular changes involve the design of tyrosine kinase inhibitors for certain patient subsets, including chronic myelogenous leukemia, lung cancer, and melanoma patients with tumors carrying translocated *Bcr-Abl*, mutated *EGFR*, and *BRAF*, respectively [9]. Efforts to enhance the applicability of this approach have led to the design of multiple clinical trials based on molecularly relevant information. The goal is that these molecular targets might function as indicators that predict the tumor’s response to pharmacological intervention. In precision oncology trials conducted until now, a striking trend has been the extensive utilization of protein kinase inhibitors in the majority of study arms [10].

DNA-targeting drugs, including cisplatin, etoposide, topotecan, mitomycin C, and gemcitabine, which are pivotal in cancer therapy, are not evaluated within precision oncology trials, except as constants maintained across all study arms. The reason is grounded in various factors, including elevated risks and difficulties faced by clinicians and institutions when sharing additional results related to existing drugs. Moreover, it takes into account patent status and diminished profitability for the companies producing these pre-existing drugs. The efficacy of precision medicine hinges on its capability to guide suitable treatments for patient groups that may derive benefits. The widest application of this rationale is observed in the utilization of pre-existing drugs. There is no rationale for assuming that these well-established drugs would be any less suitable than newer medications for a more targeted application based on the molecular characteristics of a patient’s disease. The pre-existing drugs, through reassessment via precision medicine, may be repositioned to be more effective and less harmful.

Progress in genomics and comprehension of the molecular system’s involvement in cancer has given rise to targeted therapies designed for specific molecular changes or biological characteristics. Genomics has revealed the intricate nature of cancer as a complex disease, prompting a shift in treatment approaches. The emphasis is transitioning from tumor types to individualized, genome-directed treatments based on biomarker analysis for each patient. Recent treatment strategy involves (i) identifying the target genes or pathways, (ii) exploring drugs that affect these targets (enhancing or attenuating their effects), and (iii) identifying genes or pathways that predict adverse events [11,12,13].

## 5. Strategies for Incorporating Drugs That Target DNA

### 5.1. PARP Inhibitors

Extending the genetic inactivation of DNA repair genes to homologous recombination deficiencies (HRD) within tumors, such as the inactivation of *BRCA1*, *BRCA2*, or *PALB2*, leads to increased susceptibility to PARP inhibitors, cisplatin, and topoisomerase I inhibitors. The interaction of synthetic lethality has been thoroughly described in cases of *BRCA* germline mutations and homologous recombination deficiencies (HRD) when employing PARP1 inhibition [14] (Figure 1a). Activation of PARP1 occurs in response to DNA repair intermediates such as single-strand breaks (SSB), triggering the synthesis of PAR (poly-ADP-ribose) polymers. Inhibitors like olaparib, niraparib, and rucaparib effectively block the catalytic activity of PARP1, preventing auto-PARylation. Consequently, this interference disrupts the coordination of DNA repair and enhances the stability of PARP1 binding to the DNA intermediate. The immobilization of DNA-bound PARP-1 disrupts the progression of replication forks, a phenomenon known as ‘PARP trapping’, resulting in the accumulation of double-strand breaks (DSBs). In the context of *BRCA* deficiency, where DSBs remain unrepaired, the accumulated DSBs ultimately trigger apoptosis specific to cancer cells [15]. After several pre-clinical studies elucidated the mechanisms of action, PARPi quickly advanced to clinical trials. In Table 1, a summary of clinical trials completed and in progress for PARPi in the context of *BRCA* mutation is presented. The outcomes from these studies have secured FDA (Food and Drug Administration) approval for olaparib and talazoparib in metastatic HER2-negative breast cancer with *BRCA1/2* germline mutations [16,17]. Furthermore, PARP inhibition with olaparib, rucaparib, or niraparib has been approved for treating ovarian cancers with *BRCA* germline mutation [18,19]. Maintenance therapy with PARPi (niraparib, olaparib, rucaparib) was demonstrated to substantially enhance progression-free survival (PFS) in platinum-sensitive sporadic epithelial ovarian cancer [18,20,21]. Olaparib has been approved by the FDA for metastatic castration-resistant prostate cancer (mCRPC) characterized by HRD, including *BRCA* mutation. Similarly, rucaparib is approved for mCRPC, specifically in the presence of a *BRCA* mutation [22].

PARP inhibitors, effective in specific contexts, face challenges in clinical use. Issues include resistance development, limited applicability to specific genetic mutations, potential toxicity, high treatment costs, and the ongoing need for optimal combination strategies, highlighting obstacles in their widespread therapeutic adoption across various cancers.

### 5.2. SLFN11

The expression of SLFN11 is emerging as a promising predictive biomarker for sensitivity to DNA-targeted drugs, as indicated by cell line data. SLFN11, originating from the German word “schlafen” meaning sleeping, has recently been causally linked to irreversible cell cycle arrest triggered by various DNA replication inhibitors [39] (Figure 1b). Elevated *SLFN11* expression stands out as the major factor associated with responsiveness to DNA-damaging drugs, encompassing topoisomerase I and II inhibitors, alkylating agents, and DNA synthesis inhibitors. On the other hand, the absence of SLFN11 has been linked to resistance to a wide range of DNA-damaging agents such as fluoroindenoisoquinolines, nanoliposomal irinotecan, trabectedin, and platinum drugs, as well as PARPi [39,40,41,42,43,44,45,46]. Following DNA damage, SLFN11 triggers early S-phase arrest and cell death, in contrast to SLFN11-deficient cells, which exhibit a slower progression to G2-phase and a survival advantage. More precisely, it is hypothesized that SLFN11 impedes replication by modifying the chromatin structure of replication forks following the ATR-mediated replication stress response. This suggests that interaction functions enhance the stability of paused replication forks within the intra-S and G2/M DNA damage checkpoints while also inhibiting the activation of additional replication origins [47]. High levels of SLFN11 expression have recently been associated with increased sensitivity to platinum-based chemotherapy in gastric and esophageal cancers [46,48].

In Table 2, a summary of clinical trials completed and progress for SLFN11 is presented. Some studies have demonstrated the absence of SLFN11 expression resulting from CpG promoter island hypermethylation in ovarian cancers, which is associated with diminished overall survival in patients undergoing cisplatin and carboplatin treatment [49]. While elevated SLFN11 levels in small cell lung cancer cells were associated with sensitivity to PARPi, this correlation was more pronounced, particularly with the highly potent PARP trapper talazoparib [50]. In the recently initiated Phase 2 randomized trial investigating the combination of maintenance atezolizumab with talazoparib versus atezolizumab alone for patients with SLFN11-positive extensive-stage SCLC (ES-SCLC) (SWOG1929, NCT04334941), SLFN11 expression by IHC is clinically feasible because it can be easily assessed as positive (H score > 1) or negative, and has been found to be positive in ~50% of ES-SCLC [51]. Therefore, the prognostic significance of SLFN11 may differ across distinct PARP inhibitors available in clinical practice, depending on their level of PARP trapping. Preclinical studies indicate that talazoparib exhibits the highest observed PARP-trapping capability among these inhibitors [52].

In addition, SLFN11 has shown encouraging potential as a predictive biomarker for response in ovarian and prostate cancer [43,53]. Patients with SLFN11-positive castration-resistant prostate cancer had improved radiographical PFS and prostate-specific antigen (PSA) tumor marker responses compared with patients without SLFN11 overexpression.

SLFN11, linked to drug sensitivity, faces clinical limitations. It exhibits tumor-specific effects, unclear mechanisms, and challenges with heterogeneity. Resistance development, biomarker issues, and limited trials underline the need for comprehensive research to optimize its therapeutic potential across diverse cancer types.

**Table 2 ijms-25-00752-t002:** Summary of clinical trials completed and in progress for SLFN11.

Drug	Indication	Phase	Cancer Type	Clinical Trial	Efficacy	Reference
niraparib	maintenance	2	extensive disease small cell lung	NCT05718323	n/a	
talazoparib	maintenance	2	extensive disease small cell lung	NCT04334941	atezolizumab plus talazoparib 4.2 months vs. atezolizumab 2.8 months, HR 0.70	[54]
olaparib	combination	2	relapsed small cell lung	NCT04939662	n/a	
olaparib	combination	2	uterine leiomyosarcoma	NCT03880019	PFS 6.9 months	[55]

### 5.3. Methylguanine Methyltransferase (MGMT)

Methylguanine methyltransferase (MGMT), an established marker, is not widely employed in clinical practice. MGMT, classified as a DNA repair enzyme, is present in many organs throughout the body, with its expression differing from one organ and tissue to another. Expression of MGMT was reported to be lower in tumors such as gliomas, lymphomas, breast cancer, prostate cancer, and retinoblastoma, most likely related to the methylation status of its promoter region [56]. MGMT immunohistochemistry expression has revealed substantial correlations with diverse glioma grades and subtypes, including lymphomas, thymic tumors, and pituitary tumors [57,58,59,60]. MGMT is responsible for eliminating the O6-methylguanine lesions produced by temozolomide [61]. Inhibition of MGMT by several O6-guanine derivatives and related compounds has been explored and shown to enhance temozolomide-induced cytotoxicity in cancer cells [62].

The activity of the MGMT enzyme varies in different tissues, among individuals, and at different stages of development. Among normal tissues, the liver registers the highest MGMT enzyme activity, while the lowest activity is recorded in brain tissue. MGMT activity was most pronounced in liver, ovarian, and colon tumors, yet it remained notably low in gliomas, which may have contributed to the sensitivity of glioma cells to temozolomide therapy. Some investigations suggest that heightened MGMT activity could compromise the effectiveness of alkylated anticancer medications and adversely affect patient prognosis [63].

MGMT functions as a DNA repair enzyme that prevents the cross-linking of double-stranded DNA by alkylated agents, reverses the alkylation of guanine at the O6 position, repairs DNA damage induced by drugs (including alkylating agents), and contributes to resistance against alkylating drugs [64] (Figure 1c). It also plays a role in the resistance of DNA against alkylating anticancer drugs like temozolomide, a subject extensively studied to overcome these therapeutic challenges.

While some earlier clinical trials improved the therapeutic impact of temozolomide by lowering MGMT protein expression, some trials did not achieve significant clinical benefits [65]. An increasing body of research has concentrated on enhancing the sensitivity of tumors, particularly glioblastoma, to temozolomide treatment by targeting MGMT through various approaches. Exosome-mediated circWDR62 and cyanidin-3-o-glucoside were reported to promote TMZ resistance and progression in gliomas [66,67]. Other investigations have shown that the lncRNA UCA1/miR-182-5p/MGMT axis plays a role in modulating the sensitivity of glioma cells to temozolomide via the MGMT-related DNA damage pathway [67,68]. In addition to gliomas, several studies have examined various tumors, such as melanoma, lymphoma, and cervical and ovarian cancer [69,70,71,72].

Numerous studies have explored the relationship between MGMT and resistance to tumor medications, investigating the manipulation of additional upstream and downstream signaling pathways in addition to employing the previously mentioned inhibitors. A study discovered that BanxiaXiexin decoction modulates MGMT expression by affecting IL6/JAK/STAT3-mediated PDL1 activity, thereby influencing the drug sensitivity of gastric cancer cells. This introduces a fresh perspective on treating gastric cancer by inhibiting MGMT [73]. Furthermore, it has been shown that MGMT plays a role in the chemosensitivity of cisplatin in gastric cancer [74].

The expression of MGMT could influence the efficacy of chemotherapy drugs, potentially reducing their therapeutic impact. Concurrently, chemotherapy drugs can also influence MGMT expression. Multiple studies have indicated alterations in MGMT methylation or activity in patients with glioblastoma (GBM) following chemotherapy, and the expression of MGMT in some recurrent cases differed from that in the original tumor. However, the underlying mechanisms behind these phenomena and their correlation with the selectivity of chemotherapeutic drugs for cells with high MGMT expression have not been elucidated [75].

MGMT-expressing tumors may limit alkylating agent effectiveness. Variable MGMT expression levels pose challenges for precise treatment planning. Epigenetic regulation, lack of a universal biomarker, and potential normal tissue impact highlight complexities in MGMT-targeted therapy. Resistance development adds to challenges, requiring ongoing research.

### 5.4. Ataxia Telangiectasia and RAD3-Related (ATR) Kinase

The ataxia telangiectasia and RAD3-related (ATR) kinase serves as a pivotal kinase in the DNA damage response, operating in proliferative cells during DNA replication. Its role is to secure the integrity of the genome and maintain cell viability [76]. ATR becomes activated in situations of DNA replication stress caused by various genotoxic challenges that result in phenomena such as double-strand DNA breaks, stalling of replication forks, and single-strand DNA/double-strand DNA junctions [76,77] (Figure 1d). Diverse lesions are transformed into single-strand DNA coated with replication protein A, serving as the trigger to activate and recruit ATR to DNA damage sites. Once activated, ATR works to protect genomic integrity and guarantee replication completion through various downstream effects. These include slowing the progression of replication forks, suppressing replication origin firing, ensuring a sufficient supply of deoxynucleotides, and predominantly inducing cell-cycle arrest through activation of the S–G2–M cell-cycle checkpoint. Hypomorphic ATR suppression in mice with oncogene-driven tumors has proven to be a potent inhibitor of tumor growth. These findings imply that while ATR plays a crucial role in the proliferation and survival of both normal and cancer cells, partial ATR inhibition may offer a promising avenue for anticancer therapy, ensuring a therapeutic window for normal tissues [78,79].

As a potent and selective ATR kinase inhibitor in the low-nanomolar range, BAY 1895344 demonstrates significant antitumor effectiveness in preclinical studies. When employed as a monotherapy, it exhibits activity in models featuring distinct DNA damage response (DDR) defects or oncogenic mutations that trigger replication stress. Such models include those for ovarian, prostate, and colorectal tumors and lymphomas [80]. In vivo studies have established a dose-dependent antitumor response, aligning with BAY 1895344 plasma exposure and an increase in DNA damage [81]. A randomized phase 2 clinical trial of patients with small-cell lung cancer treated with ATR inhibitor berzosertib plus topotecan did not improve progression-free survival compared with topotecan therapy alone among patients with relapsed SCLC. However, the combination treatment significantly improved overall survival [82].

Kinase inhibitors, including those targeting ATR, face challenges like off-target effects, resistance development, and potential toxicity. Patient stratification and optimizing combination strategies are critical. Limited clinical data and achieving CNS penetration add complexity, underscoring the need for ongoing research and comprehensive clinical trials.

### 5.5. Binding Strength between Anticancer Drugs and DNA

The affinity between anticancer drugs and DNA significantly influences the biological activity of these drugs. The development of an effective antitumor drug necessitates adjustments to physicochemical properties, such as lipophilicity and base strength. These modifications are critical for protein binding and metabolism, alongside the optimization of DNA-binding affinity and binding kinetics. Therefore, investigating the interaction between drugs and DNA can provide a reliable approach for screening drugs using DNA probes. Several screening assays that use different instruments have been documented, offering a straightforward approach to assessing anti-drugs in vitro. Recently developed metal nanoclusters constitute a fresh class of fluorescent nanomaterials, capturing significant interest among researchers owing to their notable characteristics such as low toxicity, high fluorescent yield, excellent photochemical stability, and compact sizes—especially silver nanoclusters [83,84].

### 5.6. Targeting DNA Hypermethylation

The strategic use of nucleoside analogs to address DNA hypermethylation has proven to be an efficient method for restructuring the epigenome of cancer cells. This intervention leads to a decrease in proliferation, better differentiation, enhanced recognition by the immune system, and, ultimately, the death of cancer cells. DNA methyltransferase inhibitors have been granted approval for the management of myelodysplastic syndromes, chronic myelomonocytic leukemia, and acute myelogenous leukemia. To enhance clinical outcomes and counteract drug resistance mechanisms, a second generation of DNA methyltransferase inhibitors has been formulated and is currently undergoing clinical trials. While effective as monotherapy for hematologic malignancies, the capacity of DNA methyltransferase inhibitors to collaborate with small molecules targeting chromatin or immunotherapy opens up additional possibilities for their prospective clinical use against both leukemia and solid tumors [85,86].

The epigenome collaborates with regulatory elements like transcription factors and noncoding RNAs to synchronize various biological processes, fine-tuning the expression or repression of the genome. Cellular signaling pathways and external stimuli further contribute to shaping epigenetics, yielding effects that are both transient and enduring. Recognizing the pivotal role of epigenetics in shaping cell functions, a more profound understanding of both normal and abnormal epigenetic processes is vital for unraveling the complexities of disease development and contemplating potential treatments, including those for cancer [87].

Similar to genome instability and mutations, epigenome dysregulation is widespread in cancer. Certain modifications influence cell function and play a role in oncogenic transformation. Nevertheless, the use of drugs or gene therapy to reverse these mutations can potentially restore the cancer phenotype to normal. A theory suggesting that epigenetic changes contribute to tumorigenesis has been put forward [88]. Variations in the likelihood of malignant transformation could be explained by the alteration of cellular methylation status through a particular methyltransferase. In tumor tissues, various tumor cells display diverse patterns of histone modification, either across the entire genome or in individual genes, indicating the presence of epigenetic heterogeneity at the cellular level [89]. Similarly, the use of molecular biomarkers is deemed a potential strategy for classifying patients into different groups. It is essential to recognize that tumorigenesis arises from the cumulative effect of multiple epigenetic events. Therefore, epigenetics serves as a tool for investigating the potential mechanisms behind cancer phenotypes and offers a spectrum of potential therapeutic options.

In contrast to genetic mutations, epigenetic alterations are reversible. With the recognition of the crucial role of epigenetic marks in tumorigenesis, inhibitors targeting these marks have attracted considerable attention. Moreover, the typical regulation of a gene through epigenetics involves multiple events. To date, hundreds of clinical trials have investigated the effects of anti-DNA methylation therapy for various cancers [86].

## 6. Conclusions

DNA-targeted drugs play a significant role in cancer treatment, offering therapeutic options for a range of diseases. Understanding DNA structure and cellular processes allows researchers to develop drugs that can precisely target and manipulate DNA, paving the way for innovative treatments and improved patient outcomes. With advances in the study of synthetic lethality, DNA repair, expression regulatory mechanisms such as epigenetic modifications, and the elucidation of comprehensive activating and inhibitory factors through technologies like ctDNA analysis, it is anticipated that more efficiently effective precision medicine will be realized. Ongoing efforts aim to tackle challenges related to this approach, encompassing the intricate task of identifying pertinent molecular events and addressing the lower-than-expected frequency of such events in patients. Focusing on distinct molecules or genes implicated in the progression and metastasis of cancer cells, targeted therapy offers a more precise and less toxic alternative to conventional chemotherapy.

## Figures and Tables

**Figure 1 ijms-25-00752-f001:**
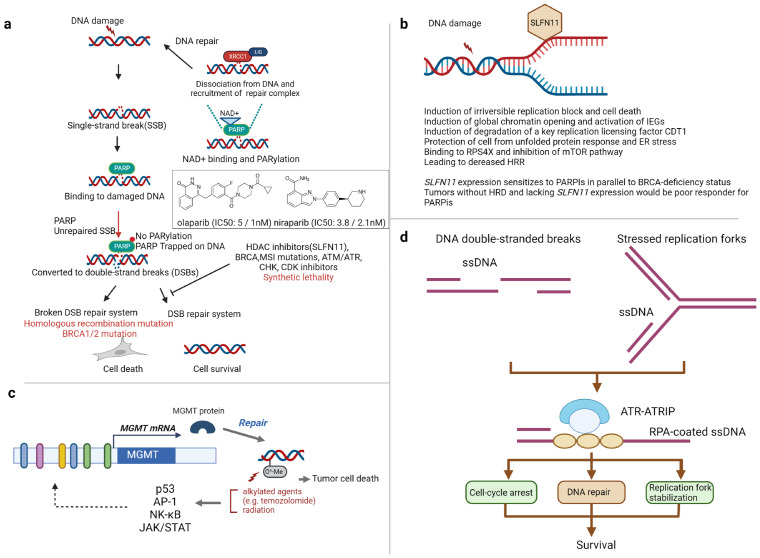
Summary of the drug targets in DNA. (**a**) Synthetic lethality: mechanism of action of PARP inhibitors. PARPs recognize damaged DNA sites and recruit DNA-repairing machinery through PARylation. Failure to repair single-stranded DNA breaks can lead to DSB, which can be precisely repaired by the homologous recombination (HR) mechanism when the DNA repair system remains intact. However, cells with *BRCA1/2* mutations progress to apoptosis. Because BRCA1/2 proteins play a key role in HR, the non-homologous end-joining (NHEJ) pathway is activated instead of HR in the case of BRCA1/2 mutated cells. Incorrect repair by NHEJ leads to genomic instability and, eventually, apoptosis. In the box are the chemical structures of PARP inhibitors with IC50 of BRCA1 and BRCA2 in cell-free assays. PARP: poly-(ADP)-ribose polymerase, SSB = single-strand DNA break, DSB = double-strand DNA break, HDAC = histone deacetylase, MSI = microsatellite instability. (**b**) Molecular mechanism of SFLN11. SLFN11 triggers a series of molecular events, as listed. When SLFN11 is absent, ATR is recruited to RPA and triggers ATR/CHK1-mediated DNA repair. Hence, the replication fork is repaired, and replication resumes. Abbreviations: IEGs immediate early genes, ER endoplasmic reticulum, HRR homologous recombination repair. (**c**) MGMT and resistance to tumor treatments, investigating the manipulation of signaling pathways, in addition to employing the previously mentioned inhibitors. MGMT functions as a DNA repair enzyme that prevents the cross-linking of double-stranded DNA by alkylated agents reverses the alkylation of guanine at the O6 position, and repairs DNA damage induced by alkylated drugs and radiation. (**d**) The ATR-kinase pathway plays a pivotal role in defending the genome from DNA damage and replication stress by overseeing and harmonizing diverse cellular processes. These processes encompass but are not restricted to, cell-cycle arrest, inhibition of replication origin firing, safeguarding stressed replication forks, and promoting DNA repair. Notably, replication protein A (RPA) is actively involved in executing these protective measures.

**Table 1 ijms-25-00752-t001:** Summary of clinical trials completed and in progress for PARPi in the context of BRCA mutation.

Drug	Indication	Phase	Cancer Type	Clinical Trial	Efficacy	Reference
olaparib	maintenance	3	ovary	SOLO2/ENGOT-Ov21, NCT01874353	PFS 19.1 months vs. placebo 5.5 months, HR 0.30	[23]
olaparib	maintenance	3	ovary	SOLO1/GOG 3004, NCT01844986	PF ratio at 3 years 60% vs. placebo 27%, HR 0.30 OS ratio at 7 years 67% vs. placebo 46.5%, HR 0.55	[24,25]
olaparib	maintenance	3	ovary	NCT03534453	PFS 16.1 m	[26]
niraparib	maintenance	3	ovary	ENGOT-OV16/NOVA, NCT01847274 (gBRCA Cohort)	PFS 21.0 months vs. placebo 5.5 months, HR 0.27	[18]
niraparib	maintenance	3	ovary	NCT02655016	PFS 21.9 months vs. placebo 10.4 months, HR 0.43	[27]
rucaparib	maintenance	3	ovary	ARIEL3, NCT01968213 (BRCA-Mutant cohort)	PFS 16.6 months vs. placebo 5.4 months, HR 0.23	[28]
rucaparib	monotherapy	3	ovary fallopian tube, or primary peritoneal	ARIEL4, NCT02855944 (Efficacy Population)	PFS 7.4 months vs. chemo 5.7 months, HR 0.64	[19]
rucaparib	monotherapy	3	ovary	NCT03522246	PFS 28.7 months vs. placebo 11.3 months, HR 0.47	[29]
Olaparib	monotherapy	3	breast	OlympiAD, NCT02000622	PFS 7.0 months vs. chemo 4.2 months, HR 0.58	[16]
olaparib	maintenance	3	breast	OlympiA, NCT02032823	PF at 3 years 85.9% vs. placebo 77.1%, HR 0.58 OS at 4 years 89.8% vs. placebo 86.4%, HR 0.68	[30,31]
olaparib	neoadjuvant	2/3	breast	NCT03150576	n/a	[32]
olaparib	monotherapy	3	breast	NCT03286842	PFS 8.11 months	[33]
talazoparib	monotherapy	3	breast	EMBRACA, NCT01945775	PFS 8.6 months vs. chemo 5.6 months, HR 0.54	[17]
niraparib	monotherapy	3	breast	NCT01905592	PFS 4.1 months vs. chemo 3.1 months, HR 0.96(censored)	[34]
niraparib	monotherapy	3	breast	NCT04915755	n/a	[35]
olaparib	monotherapy	3	prostate	PROfound, NCT02987543 (Cohort A)	PFS 7.4 months vs. control 3.6 months, HR 0.34	[22]
talazoparib	combination	3	prostate	NCT04821622	n/a	[36]
olaparib	maintenance	3	pancreas	POLO, NCT02184195	PFS 7.4 months vs. 3.8 months, HR 0.53	[37]
niraparib	maintenance	3	small cell lung	NCT03516084	PFS 1.54 months vs. placebo 1.36 months, HR 0.66(censored)	[38]

## Data Availability

No new data were created or analyzed in this study. Data sharing is not applicable to this article.

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
