# Peer review of "Biology and Development of DNA-Targeted Drugs, Focusing on Synthetic Lethality, DNA Repair, and Epigenetic Modifications for Cancer: A Review"

_ijms, 2024, doi:10.3390/ijms25020752_

Round 1
Reviewer 1 Report
Comments and Suggestions for Authors
The authors have chosen a very broad title for their review article. The authors should highlight the novelty of their topic in the abstract and introduction.
The authors should give a rationale for choosing the specific class of drugs. What is the connection between these drugs?
The title of the article is not justified appropriately. It should give an appropriate idea about the content of the article.
Some schematic figures on the mode of action of these drugs would have been attractive to the readers.
The authors should add their shortcomings in clinical therapeutic usage for each drug.
Author Response
Reply from Authors:
Thank you very much for providing important comments. We are thankful for the time and energy you expended. Our responses to the referees’ comments are as follow:
1.The authors have chosen a very broad title for their review article. The authors should highlight the novelty of their topic in the abstract and introduction.
Authors reply #1) Thank you for your comments, we focused on novelty of our topic and added perspective of the evolution of DNA-targeted drugs especially addressing synthetic lethality, DNA repair, and epigenetic modification in Title, Abstract (Page 1, line 16-21 on revised version), and Introduction (Page 2, line 48-50).
2.The authors should give a rationale for choosing the specific class of drugs. What is the connection between these drugs?
Authors reply #2) Thank you for your advice, we focused on new generation of DNA-targeted drugs, exploring synthetic lethality, DNA repair, and expression regulatory mechanisms including epigenetic modifications. This is because they are specific and effective, serve as candidates for precision medicine, and are currently under evaluation in clinical trials. We added in Introduction (Page 2, line 48-50) and 3. Characterizing cancers to targeted treatments (Page 2, line 89-92)
3.The title of the article is not justified appropriately. It should give an appropriate idea about the content of the article.
Authors reply #3) Thank you for your comments, we focused on novelty of our topic and added perspective of the evolution of DNA-targeted drugs especially addressing synthetic lethality, DNA repair, and epigenetic modification in Title of the article.
4.Some schematic figures on the mode of action of these drugs would have been attractive to the readers.
Authors reply #4) Thanks for your comments, we added graphical presentation of the drug targets in PARP inhibitors, SLFN11, MGMT, ATR-kinase including strategies targeting DNA methylation.
5.The authors should add their shortcomings in clinical therapeutic usage for each drug.
Authors reply #5) Thank you for your advice, we added summary of shortcomings in clinical therapeutic usage for each drug in PARP inhibitors (Page 4, line 154-158), SLFN11 (Page 7, line 231-234) , MGMT (Page 9, line 289-293), and ATR kinase (Page 9, line 323-327) respectively.
We thank again reviewers for their careful reading of our manuscript and their many insightful comments and suggestions.

Reviewer 2 Report
Comments and Suggestions for Authors
The review discusses the significance of DNA-targeted drugs, highlighting their role in precision medicine and their potential to tailor treatments based on individual patient genetics​​. Here are some points that need attention:
1. Expanding the introduction to include a brief historical perspective on the evolution of these drugs might enrich the context for readers.
2. A graphical representation or figure summarizing the drug targets in DNA in the review, such as PARP inhibitors, SLFN11, MGMT, ATR kinase, and strategies targeting DNA hypermethylation, would aid in visualizing the complex interactions and mechanisms of action.
3. Tables: More detailed tables with drug efficacy and patient survival data (If the data has been published) will improve the manuscript. The current one is too simple.
4. Characterizing cancers to targeted treatments. This section should be improved. It would be appreciated if the authors could add new developments in this area instead of common knowledge.
5. Enhance certain sections, such as the discussion on the affinity between anticancer drugs and DNA, by adding more clarity and detail. Elaborate on how these interactions influence the effectiveness of the medicines and outline the methods used for screening these interactions, thereby providing a deeper understanding of the subject matter.
6. The conclusion section could be strengthened by a more in-depth discussion on the future directions of DNA-targeted drug research and the challenges that must be addressed.
Comments on the Quality of English LanguageModerate
Author Response
Reply from Authors:
Thank you very much for providing important comments. We are thankful for the time and energy you expended. Our responses to the referees’ comments are as follow:
1. Expanding the introduction to include a brief historical perspective on the evolution of these drugs might enrich the context for readers.
Authors reply #1) Thank you for your comments, we added historical perspective of the evolution of DNA-targeted drugs in Introduction.
2. A graphical representation or figure summarizing the drug targets in DNA in the review, such as PARP inhibitors, SLFN11, MGMT, ATR kinase, and strategies targeting DNA hypermethylation, would aid in visualizing the complex interactions and mechanisms of action.
Authors reply #2) Thanks for your comments, we added graphical presentation of the drug targets in PARP inhibitors, SLFN11, MGMT, ATR-kinase including strategies targeting DNA methylation.
3. Tables: More detailed tables with drug efficacy and patient survival data (If the data has been published) will improve the manuscript. The current one is too simple.
Authors reply #3) Thanks for your advice. We added drug efficacy and patient survival data in Table 1 and 2.
4. Characterizing cancers to targeted treatments. This section should be improved. It would be appreciated if the authors could add new developments in this area instead of common knowledge.  
Authors reply #4) Thanks for your kind suggestion. In consideration of the context of this review, rather than adding new topics related to the examination itself, we have described the potential for more personalized treatment through novel diagnostic methods in the last part of the paragraph.
5. Enhance certain sections, such as the discussion on the affinity between anticancer drugs and DNA, by adding more clarity and detail. Elaborate on how these interactions influence the effectiveness of the medicines and outline the methods used for screening these interactions, thereby providing a deeper understanding of the subject matter. 
Authors reply #5) Thanks for your suggestion. We described assessing physicochemical properties to screen interactions in DNA-binding affinity.
6. The conclusion section could be strengthened by a more in-depth discussion on the future directions of DNA-targeted drug research and the challenges that must be addressed.
Authors reply #6) Thanks for your advice. We added future directions in the elucidation of comprehensive activating and inhibitory factors through technologies like ctDNA analysis.
We thank again reviewers for their careful reading of our manuscript and their many insightful comments and suggestions.

Reviewer 3 Report
Comments and Suggestions for Authors
The article entitled “ Biology and Development of DNA-Targeted Drugs for Cancer : 2 A Review” has been evaluated, and the article needs to be rewritten to attract readers' attraction. The authors accomplished the most recent articles; however, it needs major revision before consideration
Major Concerns
I . It must be exciting and able to attract the general audience of the journal to finish the reading or at least go through the figures with a certain level of learning. This would help not only the authors but also the influence and impact of the journal as a whole. Therefore, it is imperative for the authors to drastically improve the presentation of the figures with the goal of interesting the general audience of this journal in mind, not just a few expertise in this area.
2. For example, the authors need to consider how the readers would get the most out of Figures without reading the text for details. A more challenging task would be for the authors to find ways to maintain the readers' interest after two-thirds of the figures.
3. Although the authors list the biological activity in the text, it is just too simple to include some figures. Much more detail must be provided, such as IC50, activity dose vs the control, and what derivatives work and what not working.
4. Furthermore, the authors might consider including several graphic figures of biological studies. The copyright permission is easy to obtain. This would greatly help attract the interest of the journal's general audience.
5. Authors should also include some chemical structures of PARP inhibitors.
Comments on the Quality of English LanguageModerate editing of English language required
Author Response
Reply from Authors:
Thank you very much for providing important comments. We are thankful for the time and energy you expended. Our responses to the referees’ comments are as follow:
I . It must be exciting and able to attract the general audience of the journal to finish the reading or at least go through the figures with a certain level of learning. This would help not only the authors but also the influence and impact of the journal as a whole. Therefore, it is imperative for the authors to drastically improve the presentation of the figures with the goal of interesting the general audience of this journal in mind, not just a few expertise in this area.
Authors reply #1) Thanks for your comments, we added graphical presentation of the drug targets in PARP inhibitors, SLFN11, MGMT, ATR-kinase including strategies targeting DNA methylation.
2. For example, the authors need to consider how the readers would get the most out of Figures without reading the text for details. A more challenging task would be for the authors to find ways to maintain the readers' interest after two-thirds of the figures.
Authors reply #2) Thanks for your advice, we added graphical presentations and captions for readers to understand the mechanism of action described
3. Although the authors list the biological activity in the text, it is just too simple to include some figures. Much more detail must be provided, such as IC50, activity dose vs the control, and what derivatives work and what not working.
Authors reply #3) Thank you for your suggestion, we added figures of PARP inhibitors including IC50 of BRCA 1/2 in cell free-assays.
4. Furthermore, the authors might consider including several graphic figures of biological studies. The copyright permission is easy to obtain. This would greatly help attract the interest of the journal's general audience. 
Authors reply #4) Thanks for your advice, We added graphic figures of biological mechanisms of PARP inhibitors, SLFN11, MGMT, ATR-kinase to attract the interest of readers.
5. Authors should also include some chemical structures of PARP inhibitors.
Authors reply #5) Thank you for your suggestion, we added chemical structures of PARP inhibitors of olaparib and niraparib
We thank again reviewers for their careful reading of our manuscript and their many insightful comments and suggestions.

Round 2
Reviewer 1 Report
Comments and Suggestions for Authors
The manuscript is satisfactorily revised and can be accepted for publication.
Reviewer 2 Report
Comments and Suggestions for Authors
agree
Reviewer 3 Report
Comments and Suggestions for Authors
The authors did significant revisions based on the reviewer's comments, the MS can be acceptable for publication.
Comments on the Quality of English LanguageMinor editing of English language required